# Sex differences in self-perceived employability and self-motivated strategies for learning in Polish first-year students

Agnieszka Fudali-Czyż[1], Piotr Janusz Mamcarz[2]*, Klaudia Martynowska[3], Ewa Domagała-Zyśk[3], Andrew Rothwell[4]

1 Neurocognitive Psychology, Uniwersytet Pedagogiczny w Krakowie, Cracow, Poland, 2 Emotion and Motivation Psychology, Katolicki Uniwersytet Lubelski Jana Pawła II, Lublin, Lubelskie, Poland, 3 Special Pedagogy, Katolicki Uniwersytet Lubelski Jana Pawła II, Lublin, Poland, 4 School of Business and Economics, Loughborough University, Loughborough, United Kingdom

* pmamcarz@kul.pl

## Abstract

Self-perceived employability (SPE) is defined as the ability to attain sustainable employment appropriate to one's qualification level (Rothwell 2008) and perceived as a crucial factor in university graduates' career development. Meanwhile, University students are mainly assessed through the lens of academic achievement, which depend, inter alia, on the self-motivated strategies for learning (MSL). Firstly, we tested hypothesised sex differences in SPE's and MSL's factors in a group of the first-year university students (n = 600) in a Central European context. Our analyses revealed that female students, despite their higher results in MSL's factors (self-regulation, learning strategies, intrinsic values, self-efficacy) presented lower internal SPE than male students. Secondly, we explored how much general SPE can be predicted from general MSL, taking into account sex as a moderator, finding that sex factor was not significant as a moderator. We can consider general MSL as a good predictor of general SPE in both sex groups. The results will provide evidence to support HEI curricular development and strategies for workplace attitude change to address existing sex inequalities. In addition, our findings relating to MSL will provide evidence to support the development of approaches to enhancing student employability with additional long term benefits in mental health and well-being.

## Introduction

Self-perceived employability (SPE) is a construct encompassing both the subjective perception of opportunities offered by the external labor market and the perception of oneself as being able to find a job appropriate to one's qualifications [1]. Changing social perceptions of the value of a university degree [2] may potentially harm students' SPE. SPE correlates with numerous personal and social factors [3]. Studies based on the Social Cognitive Career Theory (SCCT) showed that personal agency (e.g., self-regulated learning and self-efficacy) is vital in the career decision-making process [3, 4].

**Data Availability Statement:** Data are available from Figshare (doi.org/10.6084/m9.figshare.16590032.v1).

**Funding:** The authors received no specific funding for this work. The funders had no role in study design, data collection and analysis, decision to publish, or preparation of the manuscript.

**Competing interests:** The authors have declared that no competing interests exist.

In higher education, students are assessed mainly through their academic achievements, which depend highly on motivation orientation towards learning. Pintrich and Groot's [5] social-cognitive framework of knowledge assumes that the self—motivated strategies for learning (MSL) are situational traits and can be learned and managed by the individual. This suggests that monitoring MSL in students from their first year of study is of great importance. It allows diagnosing what attributes and abilities students bring to the university from education at an earlier level to implement appropriate tutoring activities.

Our literature review revealed that there are reports of sex differences in MSL [1, 6–15]. However, studies carried out in different countries revealed mixed results as to sex differences in SPE [16–19]. We anticipate that these conflicting results may be due to cross-cultural differences.

The aim of our research was to test the predicted sex differences in SPE and MSL in Polish first-year students. An original contribution is that we aimed to explore the possibility of predicting SPE based on MSL in women and men in a Central European context. Hence, the paper aims to identify strategies to help students and Higher Education Institutions (HEI) in a challenging graduate employment context.

We conducted our study with 600 first-year university students. We focused on first-year students as it was considered that their MSL could be improved during the later stages of their higher-level education. In the next section, we present the component factors of SPE and MSL and consider potential sex differences in each case. Following that, we introduce the predicted connection between SPE and MSL in both groups of female and male students.

## Self-Perceived Employability (SPE) factors

SPE consists of both external and internal factors [2, 16]. External SPE factors include characteristic features and overall state of the local labor market and specific qualifications [2, 20], the perceived prestige of the chosen field of study, the subjective availability of jobs in the chosen field of study, and the perceived adjustment of the competences held to the requirements of employers. Internal SPE factors include the level of professional experience, self-efficacy, coping abilities, self-management of one's career path, mastery of one' profession, the optimal use of competencies [21] and career identity [22].

## Sex differences in SPE factors

Research shows conflicting results when it comes to the existence of sex differences in SPE. On the one hand, Spanish studies indicate higher SPE of men compared to women [16, 17]. Vargas et al. [16] found that among Spanish undergraduate students (N = 1502), men possessed higher SPE than their female colleagues. Another Spanish study conducted by Cifre et al. [17] also revealed sex differences in SPE in the case of employed (N = 181) and unemployed (N = 246) Spanish young adults (i.e., below 30y.). Employed young men perceived their internal SPE with the current employer as higher than women did. In these studies, employed young men perceived that they had more employment opportunities in their current place of employment than young employed women. On the other hand, American and British studies demonstrated–respectively: no sex differences [18, 19] or sex differences in the opposite direction, i.e., employed British women presented higher SPE in contrast to employed men [20] in a profession (human resources) that had a female majority. The explanation of these conflicting results may be due to cultural factors.

According to research based on Hofstede's model [23], Poland is more similar in the collectivism-individualism dimension to Spain than the United Kingdom and the United States. Smith [24] also suggested that Poland and Spain are more collectivist societies. We suggest

that a predominance of traditional notions of the male bread-winner may translate into a lower SPE in Polish women than Polish men. From this, we may reasonably assume that the results in our study would be similar to those obtained in Spain. Hence: We predicted that Polish male students would declare higher SPE then Polish female students (H1).

## MSL factors

Pintrich and Groot [5] created the Motivated Strategies for Learning Questionnaire (MSLQ) based on self-regulated learning notion (see also [15–17]); cyclical phase model of self-regulation (see [25]). They considered self-regulated learning as having two dimensions: 1. motivation and 2. learning strategies, in which the motivators are intrinsic-extrinsic goal orientation, self-efficacy, and test anxiety. The learning strategies consist of cognitive and metacognitive strategies, learning resources, and management strategies.

The intrinsic value of learning signifies interest in and the perceived value of learning itself. Self-regulated learning includes students' metacognitive strategies for planning, monitoring, and modifying their cognition (Pintrich and Groot, 1990, p. 33). The construct of self-efficacy is defined after Bandura [26] as the individual's judgment regarding the ability to perform a task at a certain level with the required behavior to achieve targeted performance.

Some studies have emphasized that self-efficacy is one of the best predictors of students' success [5, 27, 28] along with achievement motivation [27], emotions that arise in both success and failure [20], cognition, self-regulated learning, and test anxiety [5]. Others stress the importance of the intrinsic value of learning. Compared to self-efficacy, intrinsic value along with instrumental value might be a stronger predictor of chosen self-regulated learning strategies (memorization, elaboration, and control strategies) [29]. Furthermore, it may be that even if students understand self-regulated learning strategies and possess self-efficacy in using them, they might not use them in the non-experimental, everyday context because they are not convinced about its usefulness [30].

## Sex differences in MSL factors

Research in a range of cultural contexts has repeatedly shown that female students tended to surpass male students in terms of the claimed use of motivated learning strategies [1, 6–10]. A study amongst 185 Malaysian college students (84 men and 101 women) who chose pure sciences, mathematics, or higher mathematics indicated that girls reported a markedly higher level of self-regulatory learning [9]. Al Khatib's [7] study involving 404 United Arab Emirates (UAE) college students (204 men and 200 women) indicated that the UAE female college students displayed significantly higher means of self-regulated learning than their [Agnieszka3] male counterparts. Bembenutty [11] also found that American female students showed higher self-regulation ability and outperformed men in terms of the diversity of learning strategies used in a college biology course in New York (see also [12]). Women also appear to have higher declared anxiety than men [7, 10, 13, 14]. Hence, in our study, we predicted that Polish first-year female students would also outperform men in the case of learning strategies (H2) and self-regulation (H3) [2], and would declare higher test anxiety (H4). A further study revealed that female students are more intrinsically motivated [31]. According to Velayutham et al. [15] in relation to 719 boys and 641 girls across grades 8, 9, and 10 in 5 public schools in Perth, Western Australia, the influence of task on self-regulation was statistically significant for men only. Considering motivation factors, men seem to have a higher perceived self-efficacy [10, 13, 14]. Therefore, we further predicted that Polish first-year male students in contrast to female students would gain lower scores in the intrinsic value scale (H5), at the same time getting higher scores of MSL's declared self-efficacy (H6).

## Social Cognitive Career Theory (SCCT) and the predicted relationship between SPE and MSL

SCCT highlights that the individual perspective is central to socio-cognitive mechanisms in influencing vocational interest, choices, and career achievement [3, 4]. Personal factors such as self-efficacy and outcome expectations are essential to influence individual career actions and development [4]. Chin and Shen's [32] study revealed that self-efficacy is a crucial factor in determining individual career management and how individuals build confidence and perceive their abilities in career management. On this basis, we predicted that MSL's self-efficacy would have positive relationships with the first-year university students' internal SPE (H7). Lu [33] revealed that self-regulated learning, a notable learning skill in MSL, is another individual-related factor that has the potential to augment the extent of employability. Thus, we predicted that self-regulated learning has positive relationships with the first-year University students' SPE (H8). Taking all the above into account, we assumed that MSL would be a significant predictor of SPE in women and men (H9). Moreover, taking into account the assumed sex differences in terms of both MSL and SPE, we also Investigated whether sex may be a moderating factor for the prediction of SPE based on MSL.

## Method

### Participants

The study was conducted among first-year students, in the first semester of their studies, of the humanities and social sciences faculties in a university town in Poland. We randomly sampled 600 students from the social sciences and humanities departments, including 431 women and 169 men. The number of male and female students in the study is uneven but this mirrors the student population of the country. The average age of our participants was 19.93 years (SD = 2.33), women—19.71 years (SD = 1.45), men—20.46 years (SD = 3.69).

### The Polish adaptation of the Self Perceived Employability scale (SPE)

From among a range of available measures of SPE [34, 35], we choose the self-perceived employability (SPE) scale designed for undergraduate students in the UK [1]. This particular SPE scale has been translated in a wide range of other cultural contexts where it has proved to be robust and reliable [36]. The Polish version of SPE [36] is based on Rothwell's scale [20]. The psychometric properties of the Polish version are good: Cronbach's alpha is 0.76. It consists of 6 items to which participants respond on a Likert-type scale with anchors from "strongly disagree" (1) to "strongly agree" (5). In previous studies, scale items have been grouped into two factors: internal SPE (the perceived impact of individual attributes on employer demand for individuals in a particular field) and external SPE (broadly, perceptions of the state of the external labor market and the demand for one's subject).

### The Polish adaptation of the Motivated Strategies for Learning Questionnaire (MSLQ)

To examine MSL, we utilized the MSLQ by Pintrich and Groot [5] in the Polish language adaptation [37]. The MSLQ is a self-report instrument for students from 12 years old in which respondents assess the matching of each item to their MSL on a 7-point Likert scale (where one means 'disagree entirely' and seven means 'totally agree'). The MSLQ consists of 44 items in 5 subscales measure students' motivational orientations: (1) Self-Efficacy (9 items considering the confidence that one's abilities are sufficient to succeed), (2) Intrinsic Value (9 items regarding the reason one engages in the task), (3) Test Anxiety (4 items referring to the

emotional aspect of the task), (4) Self-Regulated Learning Strategies (13 items regarding cognitive strategies that students use to learn, remember, and understand the material) and (5) Self-Regulation (9 items considering students' effort management strategies and metacognitive strategies for planning, monitoring, and modifying their cognition). The scale is for both individual and group studies. In previous studies, respondents typically have unlimited time for completion.

The MSQL has been adapted in addition to Polish in several other countries around the world (e.g., see [38, 39]). In all versions of the tool adaptation, Cronbach's $\alpha$ coefficient was above 0.70. The reliability of the Polish language adaptation of the entire questionnaire, well as its scales, measured using the Cronbach's $\alpha$ coefficient, is satisfactory (0.89 in the test, and 0.92 in the retest after 14 days) [37]. There were significant r-Pearson correlations between the scales' results obtained in the test and retest phases, which were consistent with the theoretical assumptions of the tool.

## Procedure and ethical consideration

Participants received a short initial instruction for the study and the students had to express their informed consent in writing. They then completed the questionnaires either in-person or online. The study was entirely voluntary and participants had the right to withdraw at any time. The survey followed the ethical rules of the American Psychological Association. Ethical Committee of Pedagogy Institute John Paul II Catholic University of Lublin authorized the study (approval number:2020-10-06/1).

## Statistical analyzes

All analyzes in this study were conducted using IBM SPSS 25 for Windows. The multiple imputations technique [40] was used to estimate values for 1% of missing data. We used MANOVA to test H1-H6 about sex differences at the level of SPE and MSL factors. We calculated R Pearsons's test for correlation between SPE and MSL (H7-H8) and to test the MANOVA assumption that the dependent variables would be correlated with each other [41]. In the last step, we performed Moderated Hierarchical Multiple Regression Analyses to test H9 about the possibility to predict SPE based on MSL and to investigate whether sex may be a moderating factor for the prediction of SPE based on MSL. In this last step of analyzes, we averaged separately SPE and MSL factors into general SPE and MSL. All data associate with this research is available without restrictions (CC BY 4.0) can be accessed: https://doi.org/10.6084/m9.figshare.16590032.v1.

## Results

Before conducting the MANOVA, a series of Pearson correlations were performed between all of the dependent variables in order to test the MANOVA assumption that the dependent variables would be correlated with each other in the moderate range [41]. As can be seen in Table 1, a meaningful pattern of correlations was observed amongst most of the dependent variables, suggesting the appropriateness of a MANOVA. Additionally, the Box's M value of 50.79 was associated with a $p$ value of .007, which was interpreted as non-significant based on Huberty and Petoskey's [42] guideline (i.e., $p < .005$). Thus, the covariance matrices between the groups were assumed to be equal for the purposes of the MANOVA.

A one-way multivariate analysis of variance (MANOVA) was conducted to test the hypothesis that there would be the differences between sex (male and female) and self-perceived employability and the self—motivated strategies for learning scores.

**Table 1. Descriptive statistics and correlations for study variables.**

| Variable | M | SD | 1 | 2 | 3 | 4 | 5 | 6 | 7 |
|---|---|---|---|---|---|---|---|---|---|
| 1. Self efficacy | 43.72 | 7.24 | — | | | | | | |
| 2. Intrinsic value | 48.92 | 7.07 | .59** | — | | | | | |
| 3. Test anxiety | 16.95 | 5.78 | -.24** | .03 | — | | | | |
| 4. Strategy use | 68.18 | 9.50 | .37** | .59** | .13** | — | | | |
| 5. Self regulation | 42.64 | 7.12 | .38** | .55** | -.04 | .64** | — | | |
| 6. External SPE | 13.88 | 2.93 | .23** | .36** | .10* | .26** | .28** | — | |
| 7. Internal SPE | 7.04 | 1.59 | .45** | .28** | -.24** | .13** | .02** | .17** | — |

* $< .05$.

** $< .01$.

A statistically significant MANOVA effect was obtained, Pillais' Trace = .14, $F (7, 592) = 14.69$, $p < .001$. The multivariate effect size was estimated at .148, which implies that 14.8% of the variance in the canonically derived dependent variable was accounted for 207 by sex (Table 2). The Levene's test results showed that p-value for all dependent variables is lower then 0.05, so we have to reject the null hypothesis and conclude that the error variances of dependent variable are equal across groups.

Using the Bonferroni method, each ANOVA was tested at a .025 (.05/2) alpha level. Results demonstrated that there was sufficient evidence to determine significant effect of sex on intrinsic value, test anxiety, strategy use, self-regulation and internal SPE. In case of self-efficacy F(1, 598) = 4.53, p = .034, partial eta squared = .008 and external employability F(1, 598) = 2.30, p = .129, partial eta squared = .004, we fail to reject the null hypothesis and conclude that there is no differences between males and females.

Female score statistically significantly higher than men on the scales of intrinsic value, test anxiety and strategy use. Male, on the other hand, are characterized by higher scores only on the internal employability scale (Table 3).

## Moderation

Moderated hierarchical multiple regression analyses were used to test whether gender moderated the relationship between the self—motivated strategies for learning and the self-perceived employability. Table 4 provide the results of the hierarchical multiple regression analysis to predict employability. The regressions showed that gender did not moderate the relations

**Table 2. Means, standard deviations, and one-way MANOVA.**

| Measure | Female | | Male | | F(1, 598) | p | η2 |
|---|---|---|---|---|---|---|---|
| | M | SD | M | SD | | | |
| Self efficacy | 43.33 | 7.04 | 44.72 | 7.65 | 4.53 | .034 | .008 |
| Intrinsic value | 49.36 | 6.92 | 47.80 | 7.34 | 5.93 | .015 | .010 |
| Test anxiety | 17.64 | 5.52 | 15.18 | 6.05 | 22.81 | .001 | .037 |
| Strategy use | 69.94 | 8.95 | 63.70 | 9.40 | 57.23 | .001 | .087 |
| Self regulation | 43.22 | 6.88 | 41.15 | 7.51 | 10.41 | .001 | .017 |
| External Employability | 14.00 | 2.88 | 13.59 | 3.05 | 2.30 | .129 | .004 |
| Internal Employability | 6.91 | 1.57 | 7.38 | 1.61 | 10.61 | .001 | .017 |

**Table 3. Descriptive statistics and 95%Confidence Intervals (CI) for Mean values (M) of tested variables in women and men.**

| Variable | Female | | | Male | | |
|---|---|---|---|---|---|---|
| | M | SE | CI (95%) | M | SE | CI (95%) |
| Self efficacy | 43.33 | .348 | (42.64, 44.01) | 44.72 | .556 | (43.63, 45.81) |
| Intrinsic value | 49.36 | .339 | (48.69, 50.02) | 47.80 | .542 | (46.74, 48.86) |
| Test anxiety | 17.64 | .274 | (17.10, 18.18) | 15.18 | .437 | (14.32, 16.04) |
| Strategy use | 69.94 | .438 | (69.08, 70.80) | 63.70 | .699 | (62.33, 65.07) |
| Self regulation | 43.22 | .340 | (42.56, 43.89) | 41.15 | .544 | (40.09, 42.22) |
| External | 14.00 | .141 | (13.72, 14.28) | 13.59 | .226 | (13.15, 14.04) |
| SPE | | | | | | |
| Internal | 6.91 | .076 | (6.76, 7.06) | 7.38 | .122 | (7.14, 7.62) |
| SPE | | | | | | |

between the self-motivated and employability ($\beta$ = .169, p = .563). Only strategies for learning is a predictor of employability ($\beta$ = .418, p < .000) explaining 17% of the variance.

## Discussion

The present study has evaluated the SPE and MSL among first-year Polish male and female students (H1-H6). In general, our results confirmed sex differences in both SPE and MSL. According to our prediction, we have identified a higher level of internal SPE (H1) and MSL's self-efficacy (H6) for Poles than Polish women. Our findings of sex differences in MSL's self-efficacy are in an agreement with other studies [10, 13, 14]. Spanish research also revealed advantages in perceived men's SPE over women's SPE [16, 17]. Lower internal SPE in women than men in both Poland and Spain can mirror the predominance of traditional notions of the male bread-winner in both rather collectivist societies [23]. Sex differences in the internal SPE and MSL's self-efficacy revealed in our study may be connected with data taken from Polish population research, showing that Poles' earning expectations are higher than Polish women's. In 2014, unemployed men declared average expectations of earnings as about 13–15% higher than women [43]. Moreover, data revealed that men are more likely to choose to run a business in comparison to women and also to select a full-time job more often [44]. Lower women's internal SPE and MSL's self-efficacy may be related to employers' and own expectations related to women and men's job opportunities. Employers who hold traditional values may recruit and hire men and women in line with their beliefs about what are gender-appropriate

**Table 4. Summary of hierarchical regression analysis for gender moderation of self–Motivated strategies for learning and self-perceived employability.**

| Variable | Model | 1 | | Model 2 | | | Model 3 | | |
|---|---|---|---|---|---|---|---|---|---|
| | B | SE B | $\beta$ | B | SE B | $\beta$ | B | SE B | $\beta$ |
| Strategies for Learning | .061 | .006 | .407* | | | | | | |
| Strategies for Learning | | | | .063 | .006 | .418* | | | |
| Gender | | | | .597 | .314 | .072 | | | |
| Strategies for Learning | | | | | | | .061 | .007 | .403* |
| Gender | | | | | | | -.813 | 2.45 | -.098 |
| Gender x Strategies for Learning- | | | | | | | .007 | .012 | .169 |
| R2 | | .166 | | | .171 | | | .171 | |
| ΔR2 | | .166 | | | .005 | | | .000 | |

*p < .001.

jobs [45]. From the point of view of individual perceptions, both sex beliefs about their desire for gender roles may influence their SPE and MSL's self-efficacy, first career choices and then their general approach to the labor market [46]. At the same time, we confirmed that women in comparison to men ranked higher in terms of MSL strategy use (H2), self-regulation (H3), and intrinsic values (H5), having also higher test anxiety (H4). Worldwide studies showed that female students tended to surpass male students in terms of the claimed use of motivated learning strategies [1, 6–10]. The lowered motivation to learn in men can be associated with the situation in Polish higher education, where women outnumber men [43] and social and employer perceptions of the falling value of the higher education degree itself [47]. Our results can be associated with the fact that in 2014, Polish women had higher education more often than men. Among Polish women, as much as 24% could have a university diploma against 14% of all Poles. Among all Poles with a Master's degree (10%), Polish women outnumbered men (13% against 6% among men). For the 19–24 demographic group, in 2014, there were c. 963,000 Polish students; of which c. 582,000 thousand were Polish women (60.44%) in contrast to c. 381,000 Polish males (39.56%). In 2018, only 34% of Poles got a tertiary degree compared to 54% of Polish women (OECD, 2019) [48].

Also, we found that MSL can be treated as a predictor of the SPE (H9). Our results showed that MSL is a good predictor of the SPE in both female and male students. The higher the results of the MSL, the higher the SPE in both women and men. Furthermore, all MSL factors correlated with both external and internal SPE (H8). The strongest, positive relationship we noted between internal SPE and MSL's self-efficacy (H7). In our male group of students, both factors gained higher scores than in the female group of students. In previous studies, it was pointed out that self-efficacy [32] and self-regulated learning [7] are crucial in determining a positive approach to personal career management. The importance of the individual perspective (e.g., self-regulated learning and self-efficacy) in the career decision-making process is central to the notion of Social Cognitive Career Theory [3, 4]. As Berntson and Markland [34] recognized, perceived employability is also important to subsequent health.

## Implications

If we accept the suggestion that that MSL predicts SPE, we suggest that it is essential to carry out periodic measurements, modify the approach to students, and forecast students' subsequent coping on the labor market. Students need to understand from the point of view of their well-being and future success that there is a relationship between their MSL and SPE and between SPE and career satisfaction [48]. Despite the huge proliferation of employability research around the world (see eg. [49]), aspects related to sex are under-explored. In Central and Eastern Europe (CEE) and Poland in particular this remains an under-researched area. For example, a recent McKinsey study [50] identified the existence of the gender gap and particular challenges but focused the analysis on the CEO level. By comparison, our research offers a unique contribution as it traces both the origins of the issues, as well as some potential solutions, back into embedded Polish social values and how these are perpetuated in the Polish higher education system. For younger men, intervention in the form of attitude training is needed to address the problems of motivation and persistence. Concerning the enduring inequalities facing women in the workplace and society, especially in traditionally conservative societies, work is needed to promote greater internal SPE and MSL's self-efficacy among women. It is highly recommended for educators to use female students' orientation towards reflective learning and their general willingness to participate in developmental activity reports to enhance their self-efficacy, stressing the relationship between MSL's and SPE's skills. Having reported gender inequality effects in MSL and SPE, we suggest that future research may further

explore sex differences in embedded social values and their consequences in terms of engagement and attainment in different cultural contexts.

## Study limitations

Although we found a positive relationship between students' SPE and their MSL, it ought to be noted that we used a single measure at a given point in time. Motivation in the context of student learning may change over time thus it would be interesting to conduct a longitudinal study to test whether such occurred and whether it had a significant impact on self-perceived employability (for instance a comparative analysis of the results from the first year and last year of studies). In addition, further research could involve testing a more complex study model including variables that may moderate or mediate the relation between SPE and MSL, differentiated by sex. Indeed, future research could take more sophisticated interpretations of sex, gender and identity which have been explored, for example in Spain [17] but have yet to be explored in Poland or CEE. A final limitation relates to the research sample. In the future, it would be beneficial to derive a sample of students from several universities from different countries allowing broader generalization results.

## Author Contributions

**Conceptualization:** Ewa Domagała-Zyśk.

**Data curation:** Piotr Janusz Mamcarz.

**Formal analysis:** Piotr Janusz Mamcarz.

**Investigation:** Klaudia Martynowska.

**Methodology:** Piotr Janusz Mamcarz.

**Project administration:** Agnieszka Fudali-Czyż.

**Supervision:** Andrew Rothwell.

**Visualization:** Klaudia Martynowska.

**Writing – original draft:** Agnieszka Fudali-Czyż.

**Writing – review & editing:** Ewa Domagała-Zyśk, Andrew Rothwell.

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
