## [Decision Letter · Decision Letter 0]

18 Oct 2021

PONE-D-21-29167The relationship between self - motivated strategies for learning and self-perceived employability in the first-year Polish male and female studentsPLOS ONE

Dear Dr. Mamcarz,

Thank you for submitting your manuscript to PLOS ONE. After careful consideration, we feel that it has merit but does not fully meet PLOS ONE’s publication criteria as it currently stands. Therefore, we invite you to submit a revised version of the manuscript that addresses the points raised during the review process.

We look forward to receiving your revised manuscript.

Kind regards,

César Leal-Costa, Ph. D

Academic Editor

PLOS ONE

Journal Requirements:

2. Please change "female” or "male" to "woman” or "man" as appropriate, when used as a noun (see for instance https://apastyle.apa.org/style-grammar-guidelines/bias-free-language/gender).

3. Please improve statistical reporting and refer to p-values as "p<.001" instead of "p=.000". Our statistical reporting guidelines are available at https://journals.plos.org/plosone/s/submission-guidelines#loc-statistical-reporting.

4. Please amend your current ethics statement to address the following concerns:

a) Did participants provide their written or verbal informed consent to participate in this study?

 [The funders had no role in study design, data collection and analysis, decision to publish, or preparation of the manuscript].

Reviewers' comments:

Reviewer's Responses to Questions

**Comments to the Author**

1. Is the manuscript technically sound, and do the data support the conclusions?

Reviewer #1: Yes

Reviewer #2: Partly

2. Has the statistical analysis been performed appropriately and rigorously? 

Reviewer #1: Yes

Reviewer #2: N/A

3. Have the authors made all data underlying the findings in their manuscript fully available?

Reviewer #1: No

Reviewer #2: Yes

4. Is the manuscript presented in an intelligible fashion and written in standard English?

Reviewer #1: Yes

Reviewer #2: No

5. Review Comments to the Author

Reviewer #1: PONE-D-21-29167: The relationship between self - motivated strategies for learning and self-perceived employability in the first-year Polish male and female students

The paper aims at searching the connections between females and males’ self-motivated strategies for learning (MSQL) and employability of university students (n=600) in a Central European context. Analyses revealed significant differences in the patterns of correlation between male and female’s MSQL and employability (both internal and external). Females, despite their higher results in MSQL presented lower internal employability and general self-efficacy. According to the authors, their results will provide evidence to support HEI curricular development and workplaces attitude change to minimize existing sex inequalities.

Below are my comments for possible revision.

Major Points

Results

The authors tested 10 hypotheses in total- H1 to H7 were tested using the Independent t-test, while H8 to H10 were tested using Pearson’s Correlation. However, the authors go on to present an Intercorrelation of SPE and MLSD Factors in males and females. What is the rationale for doing this?

Also, what is the rationale for correlating MSLQ general with MSLQ factors? Certainly the correlations will be high because each of these factors contribute to the MSLQ general score. The same applies to the Intercorrelation of SPE.

Again what is the justification for the ‘sex difference in intercorrelations of MSLQ factors if the authors compared gender differences on the various variables using a t-test?

Three hypothesis, H8 to 10 predicted a relationship between specific variables and yet, the authors chose to correlate all their variable of study. Why did they not focus on the three hypothesis that had been stated? This makes it seem as though they are merely ‘fishing’ for results and therefore just decided to run as many analysis as they could to be able to generate several results.

Discussion

There is very scanty information on the discussion on H8 to H10. A lot more could have been said to explain the findings. Generally, most of the findings are not well discussed. How will the author explain their own findings and what do these findings mean?

Seven hypotheses are tested to compare males and females on several variables. There are 5 hypotheses which females scored higher than males but these are not discussed. Instead, the whole focus of the Discussion and Implications is on, and centered around the results of the two hypothesis where males scored higher than females. The question is why do the authors ignore all of these other results and just focus on the findings of 2 out of 10 hypotheses?

Generally, the focus of the Discussion seems to be all about how females presented low internal employability and general self-efficacy I find the discussion to be very biased and skewed because there were several other variables that females scored higher than males but these not discussed.

Page 10 Line 304- 310: Our results indicated that women’s mental barriers might be worse than any institutional barriers to their career advancement. Hence, there is a rising need to provide female students with psychological assistance. In the case when a person experiences learned helplessness, reflected in a sense of decreased efficiency, an appropriate intervention may involve exploring coping strategies. It is crucial to include in the higher education curriculum discussions about the psychological determinants of the underrepresentation of women in managerial positions rather than structural barriers.

What is the above assertion in reference to? Which of the study variable assessed, and which finding is it in reference to? I think this is far-fetched!

Implications

The ‘implications’ of the study is lengthier than the Discussion of results and I find that quite unusual. Again there are so many findings of several hypotheses that were confirmed in favour of females which the authors ignore.

Limitations

The authors do not discuss any limitations of the study. Note that every study has some limitations.

Minor Points

Introduction

SPE’s factors

Page 2 Line 32: External SPE factors include characteristic features of a local labour market and specific qualifications. The authors should list what some of these characteristic features of a local labour market are.

Line 88- Hence, in our study, we predicted that Polish first-year female students also

outperformed males in the case of learning strategies (H2). Use the correct tense.

The abbreviation EMP is not written in full in any part of the paper. Please specify what this stands for.

Method

Subject

It will be worth explaining how the 600 students were recruited, what sampling technique was used etc. and details about the participants such as examples of their subject of study.

The term ‘Participants’ should replace ‘Subjects’ as that is the more acceptable convention.

Polish adaptation of the self-perceived Employability Scale

Page 4 Line 130- - the authors indicates 6 items on the scale but yet lists a total of 8 items consisting of 3 internal and 5 external items.

The MSLQ

Page 4 Line 150: We considered our sample as learners whose approach to learning is still closer to school students than university students. What does closer to school students mean?

Procedure and ethical consideration

The authors should specify the name of the ethics committee where approval was obtained.

Best wishes.

Reviewer #2: Dear Editor,

Thank you for considering me to review the article entitled "The relationship between self-motivated strategies for learning and self-perceived employability in Polish first-year students". The manuscript analyses the relationships between self-perceived employability and self-motivation for learning in first-year university students.

Title

I think it does not reflect well the central aim of the study, which is evident in the discussion and the implications of the results obtained. The key issue seems to be the difference between males and females.

Abstract

The abstract is confusing because at no point does it refer to self-perceived employability, focusing on the general concept of employability. On the other hand, it should refer to the fact that they are first-year students, since this is an aspect that the authors subsequently emphasise in a special way.

Introduction

The first sentence is too long and difficult to understand (lines 2-6).

Again, as in the title, more emphasis is placed on the relationships between self-perceived employability and self-motivated strategies for learning, rather than on the differences between men and women.

SPE’s factors

From line 33 to 36: sentence difficult to understand, too long.

This section gives data on the sample size of the studies that have been reviewed, which is of no interest in this case.

Line 49 to 52: the two sentences are redundant, try to unify.

In line 55, the authors write “We predicted that Polish males would declare higher SPE then Polish”. THEN is a spelling error, the correct word is THAN.

MSL’s factors

In this section, it would be advisable for authors to include more recent and highly relevant citations. For example:

1) http://dx.doi.org/10.17220/ijpes.2017.03.001

2) DOI: 10.1016/j.psicoe.2018.09.001

3) Zimmerman, B. J., & Moylan, A. R. (2009). Self-regulation: Where metacognition and motivation intersect. In D. J. Hacker, J. Dunlosky, & A. C. Graesser (Eds.), The educational psychology series. Handbook of metacognition in education (pp. 299-315). Routledge/Taylor & Francis Group.

From line 73 to 74, the authors write: Others emphasise that a student's success will depend largely on the sense of self-efficacy, as well as the emotions that arise in both success and failure. I propose: Others stress that a student's success will largely depend on the sense of self-efficacy, as well as the emotions that arise in both success and failure [20].

Male and female differences in MSL factors

In line 99 there is a spelling error: MSLQ'a

Method

In the SUBJECTS section, it would have been advisable to include other socio-demographic variables.

In line 122, when the average age of women is given, there is an error: 19.7.1years. Remove the comma.

The Polish adaptation of the Self Perceived Employability Scale (SPE)

On line 131, correct the likert scale: the authors write from strongly disagree (1) to agree (5) strongly. It should be strongly disagree (1) to strongly agree (5).

Also, the paragraph from line 132 to 142 needs to be revised, as it is too dense and confusing. It should be reworded to make it easier to read.

The MSLQ

Present the likert scale with a similar structure to the previous one, putting the numbers in brackets instead of text.

In line 150, when the authors say, "We considered our sample as learners whose approach to learning is still closer to school" they should refer to "High School". The self-regulation mechanisms linked to learning undergo important changes in adolescence with respect to the previous stage. On the other hand, I recommend that you consult two studies that highlight the false belief that university students are capable of self-regulating their learning and of being motivated to learn:

1) https://search.informit.org/doi/10.3316/informit.191347686667042

2) https://doi.org/10.18690/rei.13.Special.129-150.2020

I recommend that the authors revise the description of the MSLQ to follow the same structure, i.e., if in the first three subscales they give a brief summary of the content of the subscales, it would also be relevant to do so for the other two. On the other hand, in line 156 they should remove "and students' learning strategies".

Procedure

In line 170, I assume they mean that the students had to express their "informed" consent in writing.

In line 171, there is an extra full stop after "online".

When the authors state that the survey had the University ethical aproval, do they mean that the University ethical committee authorised the study?

Results

The results section is perhaps the weakest point of this study, I believe that the data allow for a more robust research design and a methodological approach that goes beyond a mainly exploratory analysis through Pearson correlations. Especially given the conclusions and implications of the study. For example, a MANOVA.

On the other hand, too many tables are included when in some cases it would be more interesting to present the data in the text without resorting to tables with little elaboration and which do not follow APA standards. For example, Tables 2 and 5 would be dispensable. Table 3 is poorly developed and confusing, and its title is not even adequate. In fact, I do not understand why Table 3 refers to "intercorrelations" and Table 4 to "correlation". In both cases, it would be Correlations. In the case of Table 5, in addition to being a dispensable table, the variables should have been ordered in such a way that, on the one hand, there were the dimensions of the SPE and then the MSLQ. Nor do I see the relevance of Table 6 when these results could have been better explained in the text.

Discussion & implications

I believe that the exploratory results do not allow us to reach conclusions as resounding as those collected in the implications section.

References

Review carefully APA format errors: capital letters where they do not correspond, the names and surnames of authors only with their initials, etc.

Missing or incomplete references: 1; 11; 14; 17; 19; 26; 32; 33; 43; 44; 45; 47

6. PLOS authors have the option to publish the peer review history of their article (what does this mean?). If published, this will include your full peer review and any attached files.

Reviewer #1: No

Reviewer #2: No

---

## [Author Response · Author response to Decision Letter 0]

5 Jan 2022

Our response: We hope that our manuscript fully met PLOS ONE's style requirements in its current form.

2. Please change "female" or "male" to "woman" or "man" as appropriate, when used as a noun (see for instance https://apastyle.apa.org/style-grammar-guidelines/bias-free-language/gender).

Our response: We have changed "female" or "male" to "woman" or "man" as appropriate,

when used as a noun.

3. Please improve statistical reporting and refer to p-values as "p<.001" instead of "p=.000". Our statistical reporting guidelines are available at https://journals.plos.org/plosone/s/submission-guidelines#loc-statistical-reporting.

Our response: We have corrected our statistical reporting according to PLOS ONE's statistical reporting guidelines, including the reference to p-values as "p<.001" instead of "p=.000".

4. Please amend your current ethies statement to address the following concerns: a) Did participants provide their written or verbal informed consent to participate in this study? b) If consent was verbal, please explain I) why written consent was not obtained, ii) how you documented participant consent, and iii) whether the ethies committees/IRB approved this consent procedure.

Our response: We added in the text that "students had to express their informed consent in writing" and that "The survey followed the ethical rules of the American Psychological Association (APA, 2017). Ethical Committee of Institute of Pedagogy John Paul II Catholic University of Lublin authorized the study (approval number: 2020-10-06/1)".

5. Thank you for stating the following financial disclosure [The funders had no role in study design, data collection and analysis, decision to publish, or preparation of the manuscript]. At this time, please address the following queries: a) Please clarify the sources of funding (financial or material support) for your study. List the grants or organizations that supported your study, including funding received from your institution. b) State what role the funders took in the study. If the funders had no role in your study, please state: "The funders had no role in study design, data collection and analysis, decision to publish, or preparation of the manuscript." c) If any authors received a salary from any of your funders, please state which authors and which funders. d) If you did not receive any funding for this study, please state: "The authors received no specific funding for this work." 

Our response: We would like to state that “The authors received no specific funding for this work".

Reviewers' comments:

Reviewer's Responses to Questions

Comments to the Author

1. Is the manuscript technically sound, and do the data support the conclusions?

Reviewer #1: Yes

Reviewer #2: Partly

Our response: We hope that our manuscript fully met PLOS ONE's style requirements in its current form. We have made significant corrections to the text, including the Results and Discussion sections.

2. Has the statistical analysis been performed appropriately and rigorously?

Reviewer #1: Yes

Reviewer #2: N/A

Our response: Following the advice of reviewers, for which we are very grateful, we have refined our statistical analyzes by making the statistics more advanced (we used the one-way MANOVA instead of simple t-tests; instead of the Linear Regression, we used Moderated Hierarchical Multiple Regression Analysis; see also our responses to reviewers below).

3. Have the authors made all data underlying the findings in their manuscript fully available?

The PLOS Data policy [2] requires authors to make all data underlying the findings described in their manuscript fully available without restriction, with rare exception (please refer to the Data Availability Statement in the manuscript PDF file). The data should be provided as part of the manuscript or its supporting information, or deposited to a public repository. For example, in addition to summary statistics, the data points behind means, medians and variance measures should be available. If there are restrictions on publicly sharing data—e.g. Participant privacy or use of data from a third party—those must be specified.

Reviewer #1: No

Reviewer #2: Yes

Our response: We deposited our data to a public repository no. https://doi.org/10.6084/m9.figshare.16590032.v1 and we referred to the Data Availability Statement in the manuscript file. 

4. Is the manuscript presented in an intelligible fashion and written in standard English PLOS ONE does not copyedit accepted manuscripts, so the language in submitted articles must be clear, correct, and unambiguous. Any typographical or grammatical errors should be corrected at revision, so please note any specific errors here.

Reviewer #1: Yes

Reviewer #2: No

Our response: We made double language corrections by a native speaker and Grammarly program. We took care to improve the presentation of the data and formatting of the text according to PLOS ONE's style requirements

5. Review Comments to the Author Please use the space provided to explain your answers to the questions above. You may also include additional comments for the author, including concerns about dual publication, research ethics, or publication ethics. (Please upload your review as an attachment if it exceeds 20,000 characters)

Reviewer #1:

PONE-D-21-29167: The relationship between self-motivated strategies for learning and self-perceived employability in the first-year Polish male and female students. The paper aims at searching the connections between females and males' self-motivated strategies for learning (MSQL) and employability of university students (n=600) in a Central European context. Analyses revealed significant differences in the patterns of correlation between male and female's MSQL and employability (both internal and external). Females, despite their higher results in MSQL presented lower internal employability and general self-efficacy. According to the authors, their results will provide evidence to support HEI curricular development and workplaces attitude change to minimize existing sex inequalities. Below are my comments for possible revision.

Major Points

Results

The authors tested 10 hypotheses in total- H1 to H7 were tested using the Independent t-test, while H8 to H10 were tested using Pearson's Correlation. However, the authors go on to present an Intercorrelation of SPE and MLSD Factors in males and females. What is the rationale for doing this? Also, what is the rationale for correlating MSLQ general with MSLQ factors? Certainly the correlations will be high because each of these factors contribute to the MSLQ general score. The same applies to the Intercorrelation of SPE. Again what is the justification for the 'sex difference in

intercorrelations of MSLQ factors if the authors compared gender differences on the various variables using a t-test? Three hypothesis, H8 to 10 predicted a relationship between specific

variables and yet, the authors chose to correlate all their variable of study. Why did they not focus on the three hypothesis that had been stated? This makes it seem as though they are merely 'fishing' for results and therefore just decided to run as many analysis as they could to be able to generate several results.

Our response: We would like to express our gratitude to both reviewers for their comments on the analyzes. We removed the part with Intercorrelations. We left nine out of ten hypotheses in the text. We removed one hypothesis – original H7 about a higher relationship between intrinsic value and self-regulation in men than in women - because of its weak theoretical and empirical justification. After careful consideration, we decided to calculate the multidimentional test, the one-way MANOVA instead of several independent t-tests to test sex differences in the case of SPE and MSL factors. Moreover, we calculated Moderated Hierarchical Multiple Regression Analysis instead of the Linear Regression to take into account the sex factor as a moderator in the prediction of SPE based on MSL. 

 We added into the text: "We used MANOVA to test H1-H6 about sex differences at the level of SPE and MSL factors. We calculated R Pearsons’s test for correlation between SPE and MSL (H7-H8) and to test the MANOVA assumption that the dependent variables would be correlated with each other (...). In the last step, we performed Moderated Hierarchical Multiple Regression Analyses to test H9 about the possibility to predict SPE based on MSL and to investigate whether sex may be a moderating factor for the prediction of SPE based on MSL. In this last step of analyzes, we averaged separately SPE and MSL factors into general SPE and MSL."

Discussion

There is very scanty information on the discussion on H8 to H10. A lot more could have been said to explain the findings. Generally, most of the findings are not well discussed. How will the author explain their own findings and whatdo these findings mean? Seven hypotheses are tested to compare males and females on several variables. There are 5 hypotheses which females scored higher than males but these are not discussed. Instead, the whole focus of the Discussion and Implications is on, and centered around the results of the two hypothesis where males scored higher than females. The question is why do the authors ignore all of these other results and just focus on the findings of 2 out of 10 hypotheses? Generally, the focus of the Discussion seems to be all about how females presented low internal employability and general self-efficacy I find the discussion to be very biased and skewed because there were several other variables that females scored higher than males but these not discussed.

Our response: We provided appropriate changes to the Discussion section to refer to all hypotheses. In the Discussion section as it stands, the first paragraph is devoted to discussing the results indicating a higher declared internal SPE (H1) and MSL's self-efficacy (H6) in men compared to women. The second paragraph is devoted to the higher declared assessments of women than men in terms of MSL strategy use (H2), self-regulation (H3), intrinsic values (H5), and test anxiety (H4). The last paragraph is devoted to the hypotheses H7-H9, two of which were devoted to the correlations between the SPE subscales and MSL's self-efficacy and MSL's self-regulated learning (H7-H8), and one related to SPE based on MSL prediction.

Page 10 Line 304- 310: Our results indicated that women's mental barriers might be worse than any institutional barriers to their career advancement. Hence, there is a rising need to provide female students with psychological assistance. In the case when a person experiences learned helplessness, reflected in a sense of decreased efficiency, an appropriate intervention may involve exploring coping strategies. It is crucial to include in the higher education curriculum discussions about the psychological determinants of the underrepresentation of women inmanagerial positions rather than structural barriers. What is the above assertion in reference to? Which of the study variable assessed, and which finding is it in reference to? I think this is far-fetched!

Our response: We decided to remove the sentence with 'women's mental barriers might be worse than any institutional barriers' and softened our conclusions.

Implications

The 'implications' of the study is lengthier than the Discussion of results and I find that quite unusual. Again there are so many findings of several hypotheses that were confirmed in favour of females which the authors ignore. 

Our response: We have provided appropriate changes to the Discussion section to refer to all hypotheses testing results. We extended the Discussion section and at the same time shortened the Implication section to get rid of the suggested over-interpretation.

Limitations

The authors do not discuss any limitations of the study. Note that every study has some limitations.

Our response: We have added limitations of the study at the end of the paper.

Minor Points

Introduction

SPE's factors

Page 2 Line 32: External SPE factors include characteristic features of

a local labour market and specific qualifications. The authors should list what some of these characteristic features of a local labour market are.

Our response: We have put in the text several distinctive features of a local labor market: "the perceived prestige of the chosen field of study, the subjective availability of jobs in the chosen field of study, and the perceived adjustment of the competencies held to the requirements of employers".

Line 88-Hence, in our study, we predicted that Polish first-year female students also outperformed males in the case of learning strategies (H2). Use the correct tense.

Our response: We have corrected the tense.

The abbreviation EMP is not written in full in any part of the paper. Please specify what this stands for.

Our response: We have exchanged 'EMP' for 'employability'.

Method

Subject

It will be worth explaining of how the 600 students were recruited, what sampling technique was used etc. and details about the participants such as examples of their subject of study. The term 'Participants' should replace 'Subjects' as that is the more acceptable convention.

Our response: We have put in the text an explanation of how the 600 students were recruited by pointing the sampling technique and details about our participants such as examples of their subject of study. The term 'Subjects' was replaced by 'Participants'.

Polish adaptation of the self-perceived Employability Scale Page 4 Line 130-the authors indicates 6 items on the scale but yet lists a total of 8 items consisting of 3 internal and 5 external items.

Our response: We have corrected the number of items in the text.

The MSLQ

Page 4 Line 150: We considered our sample as learners whose approach to learning is still closer to school students than university students. What does closer to school students mean?

Our response: The sentence (page 4 line 150) has been deleted.

Procedure and ethical consideration

The authors should specify the name of the ethics committee where approval was obtained.

Our response: We specified the name of the ethics committee where approval was obtained: "Ethical Committee of Institute of Pedagogy John Paul II Catholic University of Lublin authorized the study – approval number 2020-10-06/1".

Reviewer #2:

Dear Editor,

Thank you for considering me to review the article entitled "The relationship between self-motivated strategies for learning and self-perceived employability in Polish first-year students". The

manuscript analyses the relationships between self-perceived employability and self-motivation for learning in first-year university students.

Title

I think it does not reflect well the central aim of the study, which is evident in the discussion and the implications of the results obtained. The key issue seems to be the difference between males and females.

Our response: We proposed the new title so that better reflects the central aim of the study: "Sex differences in self-perceived employability and self-motivated strategies for learning in Polish first-year students".

Abstract

The abstract is confusing because at no point does it refer to self-perceived employability, focusing on the general concept of employability. On the other hand, it should refer to the fact that they

are first-year students, since this is an aspect that the authors subsequently emphasise in a special way.

Our response: We have edited the abstract to be clearer about the objectives of the study, starting from the self-perceived employability concept, referring also to the fact that our participants were first-year university students.

Introduction

The first sentence is too long and difficult to understand (lines 2-6). Again, as in the title, more emphasis is placed on the relationships between self-perceived employability and self-motivated strategies for learning, rather than on the differences between men and women.

Our response: We have edited the Introduction section. Hopefully, we make it more concise, understandable, and with more emphasis on sex differences. Our idea was to put general concepts and aims of the study at the beginning of the Introduction section and later focused on specific problems such as sex differences in SPE and MSL in the following subsections.

SPE's factors

From line 33 to 36: sentence difficult to understand, too long. This section gives data on the sample size of the studies that have been reviewed, which is of no interest in this case.

Our response: We have edited the whole section, including the text from lines 33 to 36.

Line 49 to 52: the two sentences are redundant, try to unify.

Our response: We corrected the redundancy.

In line 55, the authors write "We predicted that Polish males would declare higher SPE then Polish". THEN is a spelling error, the correct word is THAN.

Our response: We corrected the spelling error.

MSL's factors

In this section, it would be advisable for authors to include more recent and highly relevant citations. For example:

1) http://dx.doi.org/10.17220/ijpes.2017.03.001

2) DOI: 10.1016/j.psicoe.2018.09.001

3) Zimmerman, B. J., & Moylan, A. R. (2009). Self-regulation: Where metacognition and motivation intersect. In D. J. Hacker, J. Dunlosky, &A. C.Graesser (Eds.), The educational psychology series. Handbook of metacognition in education (pp. 299-315). Routledge/Taylor & Francis Group.

Our response: We took the liberty of using the sources proposed by the reviewer to enrich the section on MSL factors.

From line 73 to 74, the authors write: Others emphasise that a student's success will depend largely on the sense of self-efficacy, as well as the emotions that arise in both success and failure. I propose: Others stress that a student's success will largely depend on the sense of

self-efficacy, as well as the emotions that arise in both success and failure [20].

Our response: We have edited the whole section including lines 73 to 74.

Male and female differences in MSL factors

In line 99 there is a spelling error: MSLQ'a

Our response: We have removed the spelling error: MSLQ'a.

Method

In the SUBJECTS section, it would have been advisable to include other socio-demographic variables. In line 122, when the average age of women is given, there is an error: 19.7.1years. Remove the comma.

Our response: We have removed the comma.

The Polish adaptation of the Self Perceived Employability Scale (SPE) On line 131, correct the likert scale: the authors write from strongly disagree (1) to agree (5) strongly. It should be strongly disagree (1) to strongly agree (5).

Our response: We have corrected the Likert scale - "strongly disagree", "strongly agree".

Also, the paragraph from line 132 to 142 needs to be revised, as it is too dense and confusing. It should be reworded to make it easier to read.

Our response: We decided to delete the paragraph from lines 132 to 142 to align the description of the SPE subscales with the following description of the MSLQ subscales. Currently, in the section describing the Polish adaptation of the SPE, there is a shortened description of subscales: "In previous studies, scale items have been grouped into two factors: internal SPE (the perceived impact of individual attributes on employer demand for individuals in a particular field) and external SPE (broadly, perceptions of the state of the external labor market and the demand for one’s subject)".

The MSLQ

Present the likert scale with a similar structure to the previous one, putting the numbers in brackets instead of text.

Our response: We have put the numbers in brackets instead of text - "strongly disagree", "strongly agree".

In line 150, when the authors say, "We considered our sample as learners whose approach to learning is still closer to school "they should refer to "High School".

Our response: After consideration, we deleted the sentence with the term "school".

The self-regulation mechanisms linked to learning undergo important changes in adolescence with respect to the previous stage. On the other hand, I recommend that you consult two studies that highlight the false belief that university students are capable of self-regulating their learning and of being motivated to learn:

1) https://search.informit.org/doi/10.3316/informit.191347686667042

2) https://doi.org/10.18690/rei.13.Special.129-150.2020

Our response: We have added the following excerpt to the manuscript in its part about MSL factors: "(...) it may be that even if students understand self-regulated learning strategies and possess self-efficacy in using them, they might not use them in the non-experimental, everyday context because they are not convinced about its usefulness".

I recommend that the authors revise the description of the MSLQ to follow the same structure, i.e., if in the first three subscales they give a brief summary of the content of the subscales, it would also be relevant to do so for the other two.

Our response: We have put a brief summary of the content of all subscales of the MSLQ.

On the other hand, in line 156 they should remove "and students' learning strategies".

Our response: We removed the indicated part of the sentence.

Procedure

In line 170, I assume they mean that the students had to express their "informed" consent in writing.

Our response: We corrected that sentence.

In line 171, there is an extra full stop after "online".

Our response: We removed an extra full stop after "online".

When the authors state that the survey had the University ethical aproval, do they mean that the University ethical committee authorised the study?

Our response: We added in the text that: "Ethical Committee of Institute of Pedagogy John Paul II Catholic University of Lublin authorized the study (approval number: 2020-10-06/1)".

Results

The results section is perhaps the weakest point of this study, I believe that the data allow for a more robust research design and methodological approach that goes beyond a mainly exploratory analysis

through Pearson correlations. Especially given the conclusions and implications of the study. For example, a MANOVA.

Our response: We would like to express our gratitude for both reviewers' comments on the analyzes. After careful consideration, we decided to calculate the multidimentional test, the one-way MANOVA. Moreover, we calculated Moderated Hierarchical Multiple Regression Analysis instead of the Linear Regression to take into account the sex factor as a moderator. Now our analyzes are unified, by taking into account the sex factor. We added into the text: "We used MANOVA to test H1-H6 about sex differences at the level of SPE and MSL factors. We calculated R Pearsons’s test for correlation between SPE and MSL (H7-H8) and to test the MANOVA assumption that the dependent variables would be correlated with each other (...). In the last step, we performed Moderated Hierarchical Multiple Regression Analyses to test H9 about the possibility to predict SPE based on MSL and to investigate whether sex may be a moderating factor for the prediction of SPE based on MSL. In this last step of analyzes, we averaged separately SPE and MSL factors into general SPE and MSL."

On the other hand, too many tables are included when in some cases it would be more interesting to present the data in the text without resorting to tables with little elaboration and which do not follow APA standards. For example, Tables 2 and 5 would be dispensable. Table 3 is poorly developed and confusing, and its title is not even adequate. In fact, I do not understand why Table 3 refers to "intercorrelations" and Table 4 to "correlation". In both cases, it would be Correlations. In the case of Table 5, in addition to being a dispensable table, the variables should have been ordered in such a way that, on the one hand, there were the dimensions of the MSLQ and then the SPE. Nor do I see the relevance of Table 6 when these results could have been better explained in the text.

Our response: We have completely rebuilt the results section, deciding to turn the stats into a more advanced one. We have included now four tables in APA style that, in our opinion, make reading the results easier.

Discussion & implications

I believe that the exploratory results do not allow us to reach conclusions as resounding as those collected in the implications section.

Our response: We have shortened the implication to get rid of the suggested over-interpretation.

References

Review carefully APA format errors: capital letters where they do not correspond, the names and surnames of authors only with their initials, etc. Missing or incomplete references: 1; 11; 14; 17; 19; 26; 32; 33; 43; 44; 45; 47

Our response: We have corrected missing or incomplete references.

---

## [Decision Letter · Decision Letter 1]

18 Feb 2022

Sex differences in self-perceived employability and self-motivated strategies for learning in Polish first-year students

PONE-D-21-29167R1

Dear Dr. Mamcarz,

We’re pleased to inform you that your manuscript has been judged scientifically suitable for publication and will be formally accepted for publication once it meets all outstanding technical requirements.

Kind regards,

César Leal-Costa, Ph. D

Academic Editor

PLOS ONE

Additional Editor Comments (optional):

Reviewers' comments:

Reviewer's Responses to Questions

**Comments to the Author**

1. If the authors have adequately addressed your comments raised in a previous round of review and you feel that this manuscript is now acceptable for publication, you may indicate that here to bypass the “Comments to the Author” section, enter your conflict of interest statement in the “Confidential to Editor” section, and submit your "Accept" recommendation.

Reviewer #1: All comments have been addressed

2. Is the manuscript technically sound, and do the data support the conclusions?

Reviewer #1: Yes

3. Has the statistical analysis been performed appropriately and rigorously? 

Reviewer #1: Yes

4. Have the authors made all data underlying the findings in their manuscript fully available?

Reviewer #1: Yes

5. Is the manuscript presented in an intelligible fashion and written in standard English?

Reviewer #1: Yes

6. Review Comments to the Author

Reviewer #1: The authors have duly addressed the comments I raised previously.

I am happy to see they have performed a MANOVA instead of the several T-tests. The results are now easier to read and follow.

I have no further comments besides the minor one below.

Limitations

“A final limitation relates to the research sample.”

The authors should specify what this limitation with the research sample is.

Ideally, the limitations are presented after the Discussion and before the Implication.

7. PLOS authors have the option to publish the peer review history of their article (what does this mean?). If published, this will include your full peer review and any attached files.

Reviewer #1: No

---

## [Editor Report · Acceptance letter]

24 Feb 2022

PONE-D-21-29167R1 

Sex differences in self-perceived employability and self-motivated strategies for learning in Polish first-year students 

Dear Dr. Mamcarz:

I'm pleased to inform you that your manuscript has been deemed suitable for publication in PLOS ONE. Congratulations! Your manuscript is now with our production department. 

Kind regards, 

on behalf of

Dr. César Leal-Costa 

Academic Editor

PLOS ONE